# Outcome Measures Utilized to Assess the Efficacy of Telerehabilitation for Post-Stroke Rehabilitation: A Scoping Review

**DOI:** 10.3390/brainsci13121725

**Published:** 2023-12-17

**Authors:** Ardalan Shariat, Mahboubeh Ghayour Najafabadi, Noureddin Nakhostin Ansari, Albert T. Anastasio, Kian Bagheri, Gholamreza Hassanzadeh, Mahsa Farghadan

**Affiliations:** 1Department of Digital Health, School of Medicine, Tehran University of Medical Sciences, Tehran 1417613151, Iran; a-shariat@sina.tums.ac.ir (A.S.); hassanzadeh@tums.ac.ir (G.H.); 2Department of Motor Behavior, Faculty of Sport Sciences and Health, University of Tehran, Tehran 1439957131, Iran; 3Department of Physiotherapy, School of Rehabilitation, Tehran University of Medical Sciences, Tehran 141556559, Iran; nakhostin@tums.ac.ir; 4Research Center for War-Affected People, Tehran University of Medical Sciences, Tehran 1417613151, Iran; 5Department of Orthopaedic Surgery, Duke University, Durham, NC 27710, USA; albert.anastasio@duke.edu; 6School of Osteopathic Medicine, Campbell University, Lillington, NC 27546, USA; kian.bagheri@duke.edu; 7Department of Anatomy, School of Medicine, Tehran University of Medical Sciences, Tehran 1417613151, Iran; 8Department of Neuroscience and Addiction Studies, School of Advanced Technologies in Medicine, Tehran University of Medical Sciences, Tehran 1417613151, Iran; 9Department of Artificial Intelligence, Faculty of Computer Engineering, Islamic Azad University of South Tehran Branch, Tehran 4147654919, Iran; farghadanmahsa@gmail.com

**Keywords:** post-stroke, dependence variable, telerehabilitation, rehabilitation assessment, telecare

## Abstract

Introduction: Outcome measures using telerehabilitation (TR) in the context of post-stroke rehabilitation are an area of emerging research. The current review assesses the literature related to TR for patients requiring post-stroke rehabilitation. The purpose of this study is to survey the outcome measures used in TR studies and to define which parts of the International Organization of Functioning are measured in trials. Methods: TR studies were searched in Cochrane Central Register of Controlled Trials, PubMed, Embase, Scopus, Google Scholar, and Web of Science, The Cochrane Central Register of Controlled Trials (Cochrane Library), the Cumulative Index to Nursing and Allied Health Literature (CINAHL), and the Physiotherapy Evidence Database (PEDro) from 2016 to June 2023. Two reviewers individually assessed the full text. Discrepancies regarding inclusion or exclusion were resolved by an additional reviewer. Results: A total of 24 studies were included in the current review. The findings were synthesized and presented taking into account their implications within clinical practice, areas of investigation, and strategic implementation. Conclusions: The scoping review has recognized a broad range of outcome measures utilized in TR studies, shedding light on gaps in the current literature. Furthermore, this review serves as a valuable resource for researchers and end users (such as clinicians and policymakers), providing insights into the most appropriate outcome measures for TR. There is a lack of studies examining the required follow-up after TR, emphasizing the need for future research in this area.

## 1. Introduction

The use of innovative technology for the treatment of cognitive and motor impairments in stroke during the critical golden hour is of paramount importance [1]. Recently, the use of telerehabilitation (TR), which we define as the ability to provide assessment and intervention to people who require rehabilitative services via telecommunication, has emerged as a substitute for in-person therapy [2]. Recent studies have shown that TR can positively affect motor functions such as balance, mobility, and postural control [1,3].

TR offers a potential solution to some of the accessibility challenges faced by individuals living with stroke [2,4]. A study found that TR interventions for stroke found no change between telehealth and face-to-face interventions for activities of daily living, balance, and upper extremity involvement [5]. Within TR, communication between patients and qualified rehabilitation professionals is facilitated via technologies like telephones and internet-based videoconferencing. Analyzing the efficacy of these interventions is pivotal for advancing the field of TR [1]. Numerous tools have been developed to assess both the outcomes and the effectiveness of post-stroke interventions [6]. 

There is a growing need for improvements in stroke care [7]. The latter study provides strong evidence supporting the effectiveness of both virtual reality (VR) and TR in enhancing stroke care, offering valuable guidance on selecting appropriate outcome measures for assessing the effect of these interventions on survivors of stroke and their families [7]. A recent literature review recognized numerous assessment tools utilized in stroke therapy [8]. Another review of outcome measures utilized in randomized controlled trials (RCTs) identified 30 distinct measures documented in RCTs, which gauged the efficacy of interventions in stroke therapy [9]. The adequacy of TR relative to the status quo is confirmed when outcome measures demonstrate no significant decline in performance compared to traditional treatment [5]. Thus, choice of an appropriate outcome metric to utilize in research and in clinical practice is imperative. 

It is important to note that when selecting outcome measures for clinical observation for patient improvement, the consideration should assess not just impairments in motor function, but also encompass various factors such as the patient’s lifestyle and daily preferences [9]. There are numerous advantages to employing standardized outcome measures, which include the ability to identify patients at risk of experiencing adverse or unfavorable outcomes, identifying the most effective interventions tailored to specific contexts, and analyzing organizational metrics [2]. Clinicians have supported the utilization of standardized tools in therapy for several years. A study by Diana et al. in 2017 emphasized the importance of clear outcome measurements with a focus on TR and VR [10]. However, there remains a lack of consensus regarding the utilization of outcome measures to enable meaningful appraisals across interventions and studies [4]. This gap in consensus has persisted from January 2015 until the present day, especially within the realm of TR. Considering the COVID-19 pandemic, during which the healthcare industry relied heavily on telerehabilitation interventions, there is a pressing need for establishing a consistent approach in this regard [11]. In addition, using telerehabilitation is beneficial for patients who cannot commute to clinical settings, particularly in rural and isolated areas [12]. Telerehabilitation also has the potential to reduce the costs of hospitalization for some patients [13]. Peretee et al. in 2017 found that telerehabilitation is effective in caring for patients with severe pathologies, such as serious cognitive deficits, enabling them to undergo physiotherapy at home without the need for exhausting transportation [13]. TR is also well-suited for patients residing in rural areas, distant from urban clinical centers, who need rehabilitation during the critical golden hour [13]. Virtual reality serves as a technology for home-based rehabilitation, providing a safe environment for patients to engage in conventional exercises, even though some studies explore the application of TR in virtual environments [14].

The current review is the first to our knowledge that attempts to elucidate the outcome measures employed in the rapidly evolving field of TR. The most recent telerehabilitation technologies include exergames (e.g., the XR-MoBI technology), digital applications, digital health technologies, telecommunication methods, and mobile applications used as treatments for patients [15].

With the aim of establishing comprehensive guidelines for the utilization of outcome measures in TR, particularly within the realm of stroke rehabilitation, we have conducted a scoping review that systematically synthesizes the prevalent outcome measurement practices. Thus, the present study aims to delineate the findings from this scoping review.

## 2. Methods

The Arksey and O’Malley framework from the University of York was used as guidance for a methodologically rigorous approach to systematically review the outcome metrics utilized to evaluate the efficacy of TR [16]. The York framework has been used broadly in knowledge synthesis trials and consists of the following five stages: (1) classifying the research question; (2) recognizing pertinent studies based on the research question; (3) trial selection; (4) charting the information within the selected trials; and (5) organizing, summarizing, and reporting the findings of the scoping review. The research questions for the current review were as follows: which outcome measures are used in TR stroke therapy trials and at what time points are they controlled (admission, discharge, and follow-up of the patient) subsequent to a stroke? Which functions from the International Classification of Functioning (ICF) are assessed in the outcome measures? This study was carried out in accordance with the Preferred Reporting Items for Systematic Reviews and Meta-Analyses (PRISMA) guidelines [17].

### 2.1. Eligibility Criteria

The inclusion criteria for this scoping review consisted of trials: (1) including patients that had sustained a stroke, (2) recounting a rehabilitation protocol utilizing TR, (3) written in English, and (4) published after January 2015. The exclusion criteria included: (1) non-English manuscripts, (2) papers omitting outcome measures, (3) papers only reporting laboratory measures, (4) discussion and protocol papers or commentary and qualitative studies, (5) poster presentations, abstracts, or papers lacking information about the treatment, and (6) papers only reporting the change and development of the technology. The search was completed using study design or publication date.

### 2.2. Search Strategy

The literature search was done by a librarian in the field of therapy. The search included PubMed, Embase, Scopus, Google Scholar, Web of Science, The Cochrane Central Register of Controlled Trials (Cochrane Library), the Cumulative Index to Nursing and Allied Health Literature (CINAHL), and the Physiotherapy Evidence Database (PEDro) (until July 2023) to classify potentially related studies.

### 2.3. Data Collection Process

Two of the reviewers (MGN and MF) independently investigated the titles and abstracts extracted from the database searches to determine if they fit the inclusion criteria. Disagreements regarding the inclusion or exclusion of a particular manuscript based on the appraisal of its abstract were determined by reaching an agreement or consulting an additional reviewer (AS). Data extraction arrangements were established based on the current literature in the field and on the questions of the research. Extraction of the data was based on essential information according to questions of the current review such as (a) the study’s authors, (b) the publication date, (c) the objective(s) of the trial, (d) the design of the trial, (e) country, (f) outcome measures reported, (g) patient characteristics (e.g., age, sex, socioeconomic status, level of education, motor functional level, the phase of the stroke, type of the stroke), (h) related ICF domains, (i) period of time at which the assessment was taken (e.g., admission, discharge, follow-up), (j) technology used for TR, and (k) details on the TR intervention. The outcome measures were categorized based on the ICF domains [16].

### 2.4. Critical Appraisal of the Included Articles

The modified Critical Appraisal Skills Programme (CASP) tool [18,19] was used for assessing the quality of each of the included studies by the three reviewers (MGN, MF, and AS). The CASP tool is an instrument used for evaluating the strengths and limitations of any qualitative research approach [19]. The tool has 10 questions that each emphasizes different methodological domains of a qualitative study: the identification of the research questions, the relevance of the methodology (including study design), description of the population and sample size, outcomes, suitability of analysis methodologies, relevance, and clarification of results. Information was obtained from studies achieving scores greater than 50% based on the CASP scoring system.

### 2.5. Quality Assessment

We used the CASP tools for assessing the quality of studies, primarily case-control studies and clinical trials. The CASP RCT checklist evaluates 11 critical criteria:(1)Did the study address a clearly focused research question?(2)Was the assignment of participants to interventions randomized?(3)Were all participants who entered the study accounted for at its conclusion?(4)Was blinding appropriately addressed for participants, assessors, and therapists?(5)Were the study groups similar at the start of the randomized controlled trial?(6)Apart from the experimental intervention, did each study group receive the same level of care (i.e., were they treated equally)?(7)Were the effects of intervention reported comprehensively?(8)Was the precision of the estimate of the intervention or treatment effect reported?(9)Did the benefits of the experimental intervention outweigh the harms and costs?(10)Could the results be applied to your local population/in your context?(11)Would the experimental intervention provide greater value to the people in your care than any of the existing interventions?

The CASP case-control study checklist also consists of 11 questions:(1)Did the study address a clearly focused issue?(2)Did the authors use an appropriate method to answer their question?(3)Were the cases recruited appropriately?(4)Were the controls selected appropriately?(5)Was the exposure accurately measured to minimize bias?(6)Aside from the experimental intervention, were the groups treated equally, and did the authors account for the potential confounding factors in the design and/or in their analysis?(7)How large was the treatment effect?(8)How precise was the estimate of the treatment effect?(9)Are the results credible?(10)Can the results be applied to the local population?(11)Do the results of this study fit with other available evidence?

Responses to these questions were recorded as ‘‘Yes”, “No”, or “Can’t tell”. In the current review, seven studies were evaluated using the CASP RCT checklist [20,21,22,23,24,25,26] (Table 1).

In addition, when appraising other studies using the CASP case-control study checklist, questions 4 (Were the controls selected appropriately?) and 6 (Aside from the experimental intervention, were the groups treated equally, and did the authors account for the potential confounding factors in the design and/or in their analysis?) were deemed not applicable since the reported trials were uncontrolled trials. Thus, the total number of questions for the latter studies was nine rather than 11. Sixteen out of the 23 trials had scores between 7 and 9 out of 9, with only two studies scoring 7. Six of the included trials had a score between 8 and 9 out of 11, whereas only four studies scored 7.

## 3. Results

The exploration of the electronic databases recognized 550 manuscripts after duplicate studies were removed. After screening of the titles and abstracts, 136 studies remained. After a full-text review process, 110 articles were excluded, leaving a total of 24 included studies. Reasons for exclusion of studies are depicted in Figure 1.

### 3.1. Included Studies

The current scoping review encompassed a comprehensive analysis of 24 studies. This review is organized into three key sections: (a) essential characteristics of the trials, which include details about the authors, location, publication year, study design, subject characteristics, type of stroke, TR explanation, and the numerical score of the quality of the studies above 7 from 9 related to pooled studies, (b) TR outcome measures used in assessing post-stroke patients, and (c) areas of the ICF covered by these outcome measures. The included trials were published between 2015 and 2023, and most of the trials were conducted in the USA and Canada. The most common study designs were quantitative approaches such as RCTs, CTs, case studies with one group and two groups with pre- and post-test intervention (Table 1).

### 3.2. Participant Characteristics

The study participants primarily consisted of males (335) who had experienced various stroke conditions, including ischemic, subacute, and chronic stroke with symptoms such as hemiparesis, aphasia, and other neurological disorders. These individuals were willing and consenting to begin a rehabilitation protocol. All studies provided detailed information on age, gender distribution, and the total number of participants. Two of the studies included a single case study involving post-stroke patients (Table 1). All the studies used TR intervention and two studies used TR with VR. The TR interventions were provided via various modalities, including video games, an internet-connected computer and laptop, TR application, serious games, and robot-based TR (Table 1)

### 3.3. Frequently Used Outcome Measures

A total of 20 outcomes were used in the scoping review (15 outcomes in TR studies and 5 outcomes in TR studies with VR). The most used outcomes were the Fugel–Meyer assessment of the recovery of patients with stroke (FMA) [20,24,35,42], balance, and motor function in the upper limb function. All outcome measures were used pre- and post-protocol based on TR (Table 2).

### 3.4. ICF, Disability, and Health Domain

The ICF serves as a framework comprising domains or categories, offering valuable guidelines for reporting functioning, performance, and health in clinical assessments. In the current study, none of the trials employed the ICF guidelines for outcome measurement encompassing aspects of both upper and lower limb function, structural aspects, and physical activity. The majority of the pooled studies focused on upper limb function (trunk mobility and functional recovery) [21,23,28,32,35,42] and some studies focused on lower limb function (balance and gait) [25,26,30,32,34,38].

## 4. Discussion

In recent years, TR has emerged as a new technology for treating and rehabilitating stroke patients [34]. In the current review, we identified more than 20 outcome measures (Table 2) that illustrate a broad range of assessments utilized in trials focused on stroke rehabilitation with interventions provided through TR. Among these measures, the most used was the FMA. FMA is a performance-based deficiency index and is designed to measure motor function, balance, awareness, and joint functioning in stroke patients. It serves multiple purposes, including measuring motor recovery, assessing disease severity, and aiding in treatment planning and evaluation.

In contrast, other studies have employed various other tools to assess a common outcome such as balance [7]. These tools encompass diverse measurements, including gait speed, Barthel Index (BI), Berg Balance Scale (BBS), Stroke Impact Scale (SIS), and quality of life (QOL) metrics. Importantly, the FMA has demonstrated outstanding reliability in both inter-rater and intra-rater assessments, exhibits strong construct validity, and is highly responsive to detecting changes in patient’s conditions. The intraclass correlation coefficient (ICC) for both the intra- and inter-rater reliability of the FMA both had values above 0.90, consistent with the reliability of this tool for stroke in the chronic and subacute phases. For validation of measuring the strength of association, the ICC and other correlation methods are necessary.

The BBS is another reliable tool, but it is not sensitive enough to detect subtle yet clinically significant changes in balance in individual subjects, particularly those recovering from stroke [15]. It is a relatively inexpensive test and can be used with a wide range of populations, including healthy individuals and patients. It evaluates balance through a comprehensive assessment that encompasses two distinct dimensions, static and dynamic, via a structured questionnaire [43,44].

Gait analysis is another valuable measurement that was utilized in five of the included studies to meticulously assess details of step and gait speed in stroke patients [34]. In addition, the Stroke Impact Scale (SIS) is a widely used measure due to its reliability, validity, and sensitivity to change [45]. The SIS contains a question to evaluate the patient’s global perception of their percentage of recovery [46]. Another frequently utilized measure that was used in studies is the Barthel Index (BI). However, there is a strong need for greater consistency in methods, content, and scoring across studies, given that the “BI” acronym is associated with various assessment methodologies. For example, some studies have adopted a 10-item scale, scoring on a range of 0 to 100 with 5-point increments [47]. This approach has been used in several multicenter stroke trials, and we call for more uniform application of this tool for stroke trials. Consistency in result reporting will allow for more appropriate pooling of data for literature review and meta-analysis.

In general, all the aforementioned outcome measures aim to capture important changes in patients who are undergoing stroke rehabilitation, whether by TR or more traditional means. Importantly, most studies have highlighted that patient satisfaction plays a pivotal role in their recovery and motivation to continue with rehabilitation to regain function. Surprisingly, only two studies incorporated a thorough assessment of patient satisfaction and motivation, using tools including the Client Satisfaction Scale (CSS) and the Canadian Occupational Performance Measure (COPM). Upon examining the satisfaction levels of patients who underwent TR following a stroke, the results unequivocally indicate that TR can be a highly effective intervention in the realm of rehabilitation. A study even mentioned maintenance of long exercises in telerehabilitation as feasible; ultimately telerehabilitation can prevent deterioration, improve physical performance, health status, and quality of life [41].

Our scoping review identified various evaluation questions that pertained to changes in health service utilization, intervention costs, and the utilization of comprehensive assessment tools to gauge aspects of patient safety, comfort, ease of use, and the efficiency-related consequences resulting from interactions with the technology [48]. This scoping review focused on motor functions such as upper-extremity function, balance, and postural control, yielding outcomes similar to those observed in previous research, such as the study conducted in 2017 [7]. Notably, the trial of Tate et al. found a limited number of studies (8.8%) that assessed specific motor, sensory, and other bodily functions [47]. It is worth mentioning that most of the studies reviewed in this study predominantly evaluated domains related to mental function [47]. In contrast, our scoping review identified only two studies that used the Mini Mental State Examination (MMSE). Future studies should prioritize outcome measures that support ICF domains using TR. Adhering to the Canadian Best Practice Recommendations for Stroke Care can comprehensively cover the various aspects of the ICF framework during both the short- and long-term recovery in stroke patients.

## 5. Conclusions

Our review included quantitative studies such as RCTs, CTs, case studies that provided essential information regarding participant demographics, including age and sex, as well as details about the interventions and the specific type of TR employed for rehabilitation. Most of these studies assessed outcomes related to motor function, consistently reporting improvements in this domain. However, it is important to note that most studies did not include information about the cost implications of the interventions, which could provide valuable insights for healthcare providers, clinicians, patients, and their families when making decisions based on using new technology with TR. Future studies should emphasize measuring the utilization and feasibility of these outcomes within the context of TR while also providing detailed cost-related information. Furthermore, future studies should investigate the standards that guide the selection of outcomes by clinicians and investigators. Furthermore, incorporating standard exercises can facilitate the learning and correction of general motor patterns, leading to noticeable improvements. It is crucial to explore the reasons behind the exclusion of certain outcomes, such as the need to establish new protocols for professionals, ensuring the availability of assessment tools in the same language as the patients, managing the time required for assessments, and addressing equipment-related prerequisites for the utilization of specific tools. Understanding and addressing these factors will contribute to the improvement of outcome selection processes in TR and related research. Exploring comprehensive methods to assess intervention costs and investigating potential variation in TR acceptance among different demographic groups could be impactful. The development of an application for assessment based on standardized measurements is essential for telerehabilitation, as physiotherapists can monitor activation and compare movement patterns. On the other hand, future studies must further assess follow-up outcomes for TR and characterize the effect size over the long term.

## Figures and Tables

**Figure 1 brainsci-13-01725-f001:**
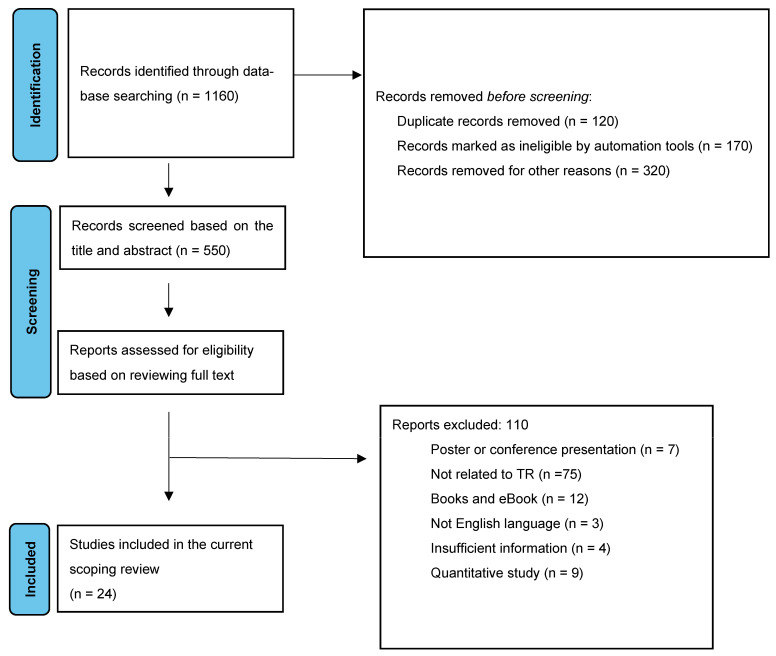
PRISMA 2023 flow diagram for the scoping review about TR and stroke as rehabilitation.

**Table 1 brainsci-13-01725-t001:** The characteristics of the included studies.

First Author, Year—Country	Design; Participant’s Age Group; Sex	Type of Stroke; Phase of Stroke Rehabilitation	Type of VR or TR Brief Description of The System	CASP
Cramer; 2023 [20]—USA	Randomized clinical trial; 124 adults; M = 90, F = 34, age of 61	Stroke with arm motor deficits	TR:	8/11
Toh; 2023 [27]—Hong Kong	Mixed-method study; 11 adults; M = 4, F = 7, age ≥ 18 years	Limb telerehabilitation in persons with stroke	TR: used wearable device, telerehabilitation application	9/9
Contrada, 2022 [28]—Italy	Clinical trial study; 19 patients M=13 F = 6; age: 61.1 ± 8.3 years	Post-stroke patients with a diagnosis of first-ever ischemic (*n* = 14) or hemorrhagic stroke (*n* = 5)	TR: The entire TR intervention was performed (online and offline) using the Virtual Reality Rehabilitation System (VRRS) (Khymeia, Italy).	9/9
Allegue; 2022 [29]—Canada	Mixed-method case study; 5 adults M = 3, F = 2; age: 41–89	Stroke survivors	TR+VR: (VirTele): virtual reality combined with telerehabilitation	9/9
Salgueiro; 2022 [30]—Spain	Prospective controlled trial; 49 adults M = 31, F = 18; age: 55–82	Subjects with a worsening of their stroke symptoms or any of the comorbidities (e.g., another neurological disease or orthopedic problem of the lower limbs)	TR: using AppG	9/9
Salgueiro; 2022 [31]—Spain	Prospective, single-blinded, randomized controlled trial; 30 adults M = 20, F = 10; over 18 years of age	Chronic stroke survivors	TR: The practice of specific lumbopelvic stability exercises, known as core-stability exercises	9/11
Anderson; 2022 [32]—USA	Case study design and experimental study;, one participant F = 1; 37 years old	Stroke with the etiology was a subarachnoid hemorrhage caused by a ruptured aneurysm at the left middle cerebral artery bifurcation	TR: framework for telerehabilitation and the effects of team-based remote service delivery	9/9
So Jung Lee; 2022 [26]—Republic of Korea	Randomized control trial (RCT); 17 adults eligible; 14 participants finished M = 10, F = 4; age: experimental group = 9 control group = 8	Patients with subacute or chronic stroke	TR: videoconferencing using Zoom	8/11
Dawson; 2022 [33]—Canada	Pilot, single-blind (assessor), randomized controlled trial (RCT); 17 adults; M = 9, F = 8; age: 42–75	Stroke survivors fluent in written and spoken English and with no severe aphasia	TR: a strategy training rehabilitation approach (tele-CO-OP)	8/11
Uswatte; 2021 [21]—Birmingham	Randomized clinical trial; 24 adults ≥1-year post; age: 48–72 M = 13, F = 11	Upper-extremity hemiparesis after stroke	TR using a computer-generated random numbers table, in-lab or telehealth delivery of CIMT	8/11
Rozevink, 2021 [23]	Randomized controlled; M = 8 F = 3; age = 66.0 ± 8.4	Upper limb function after stroke	TR: home-care arm rehabilitation (MERLIN), a combination of an unactuated training device using serious games and a telerehabilitation platform in the patient’s home situation	9/9
Rozevink, 2021 [24]	Randomized controlled; M = 8 F = 4; age = 64.8 ± 8.5	Upper limb function in chronic stroke	TR: home-care arm rehabilitation (MERLIN); telerehabilitation using an unactuated device based on serious games improving the upper limb function in chronic stroke	8/9
Shih-Ching, 2021 [34]	Prospective case-controlled pilot study; 30 patients F = 6 M = 9; age: 51–68	Chronic stroke	TR: three commercially available video games	9/9
Chingyi, 2021 [35]	A single-group trial; 11 participants F = 6 M = 5; age: 44–66	chronic stroke (hemorrhagic/ischemic)	TR: home-based self-help telerehabilitation program assisted by the aforementioned EMG-driven WH-ENMS	7/9
Marin-Pard, 2021 [36]	Case study and clinical trial study; one participant M = 1; age = 67 years old	Chronic stroke with upper extremity hemiparesis	TR: tele-REINVENT system consisting of a laptop computer with all necessary programs preloaded, configured, and displayed in an easy-to-use manner, a pair of EMG sensors with the enclosed acquisition board, and a package of disposable electrodes	7/9
Cramer; 2021 [21]—USA	Prospective, single-group, therapeutic feasibility trial; 13 adults M = 9, F = 4; median age 61	Home-based telerehabilitation after stroke	TR: patients received 12 weeks of TR therapy, 6 days/week, with a live clinic assessment at the end of week 6 and week 12. Patients were free to call the lab with questions	9/9
Kessler; 2021 [37]—Canada	Multiple baseline single-case experimental design; 8 adults M = 6, F = 2; age: 50–83	Stroke survivors	TR: telerehabilitation occupational performance coaching	9/9
Saywell, 2020 [25]	Randomized controlled trial; ACTIV: n = 47; control: n = 48 N = 95 participants M = 49 F = 46	Participants had experienced a first-ever hemispheric stroke of hemorrhagic or ischemic origin and were discharged from inpatient, outpatient, or community physiotherapy services to live in their own home	TR: augmented community telerehabilitation intervention	9/11
Burgos; 2020 [38], Chile	Clinical study; 6 participants M = 3 F = 3	Chronic stage: in early subacute stroke (seven weeks of progress)	TR: low-cost telemedicine (therapist monitoring was carried out by connecting to the web platform and watching games scores daily at the scheduled session time or afterwards based on therapist availability)	9/9
Ora; 2020 [22]—Norway	Pilot randomized controlled trial; 30 adults; M = 19, F = 11; age > 18	Post-stroke with aphasia	TR: using a portable Fujitsu PC (laptop) with necessary software and material	9/11
Huzmeli; 2017 [12]—Turkey	Clinical trial study; 10 adults M = 6, F = 4; age: 45–60	Patients with stroke who were hemiplegic and had sufficient equipment	TR: video communication(TR was applied by contacting the patients via laptops with a camera and microphone and an internet connection)	9/9
Ivanova; 2017 [39]—Germany	Clinical trial study; 6 participants M = 4 F = 1; age: 51–89 years	Motor relearning after stroke (five patients were in the subacute phase; one patient was considered chronic. All participants showed deficits in the motor activity of the shoulder, arm, and hand function)	TR: haptic devices for stroke rehabilitation and robot-based telerehabilitation system	9/9
Dodakian; 2017 [40]—USA	Clinical trial study; 12 adults M = 6, F = 6; age: 26–75	Patients with chronic hemiparetic stroke	TR: individualized exercises and games, stroke education	9/9
Özgün; 2017 [41]—Turkey	Pilot study; 10 adults M = 6, F = 4; age = 44–61	Patients with stroke	TR: giving rehabilitation services with computer-based technologies and communication tool	8/9

**Table Legend**. VR, virtual reality; TR, telerehabilitation; CASP, cognitive assessment scale for stroke patients; AppG, access to telerehabilitation to perform core stability exercises at home; CIMT, constrained-induced movement therapy; EMG, electromyography; WH-ENMS, wrist/hand exoneuromusculoskeleton.

**Table 2 brainsci-13-01725-t002:** Frequency of used outcome measures in TR intervention studies.

Study (First Author, Year)	Standardized Outcome	Instrument	Reported Findings	ICF Domain	Focus of the Outcome
Cramer; 2023 [20]—USA	Upper and lower limb function	Fugel–Meyer motor assessment	Telerehabilitation has the potential to substantially increase access to rehabilitation therapy on a large scale	b730	Suboptimal rehabilitation therapy doses
Toh; 2023 [27]—Hong Kong	Usability of the wristwatch	System usability scale (SUS) questionnaire	Usability of the proposed wristwatch and telerehabilitation system was rated highly by the participants	S730	Upper limb
Contrada, 2022 [28]—Italy	Motor recovery	Barthel Index (BI); Fugel–Meyer motor score (FM) and Motricity Index (MI)	TR tool promotes motor and functional recovery in post-stroke patients	b730	Upper limb
Allegue; 2022 [29]—Canada	Improvement of UE motor function	Berg balance assessment functional gait assessment: activity-specific balance confidence scale independently applied	Most stroke survivors found the technology easy to use and useful	b730	Arm feasibility
Salgueiro; 2022 [30]—Spain	Balance in sitting position	The Spanish-version of the Trunk Impairment Scale 2.0 (S-TIS 2.0), Function in sitting test (S-FIST), Berg Balance Scale (BBS), Spanish-version of postural assessment for Stroke patients (S-PASS), Brunel Balance Assessment (BBA) gait assessment	Greater improvement in balance in both sitting and standing position	b730	Feasibility of core stability exercises
Salgueiro; 2022 [31]—Spain	Balance and gait	Spanish-Trunk Impairment Scale (S-TIS 2.0), sitting test, Spanish postural assessment scale	Improvement in trunk function and sitting balance	b730	Trunk control, balance, and gait
Anderson; 2022 [32]—USA	Feasibility and acceptability, satisfaction	The Canadian Occupational Performance Measure (COPM), a standardized semi-structured interview	Tele-CO-OP was found to be feasible and acceptable	b730	Feasibility and acceptability based exercise
So Jung Lee; 2022 [26]—Republic Of Korea	Trunk control and balance function, the functional movement and locomotion necessary for sitting, standing, and walking, dependent walker, ADLs, health-related QoL	Trunk Impairment Scale (TIS) scores, the Berg Balance Scale (BBS), timed up and go (TUG) test, functional ambulation categories (FAC), Korean Modified Barthel Index (K-MBI) scores EuroQoL 5 Dimension (EQ-5D) tool	Significant improvement in the TIS scores	b730	Subacute or chronic stroke
Dawson; 2022 [33]—Canada	Self-identified in everyday life activities and mood	Canadian Occupational Performance Measure (COPM), the PHQ-9	High satisfaction and engagement	b730	Improvements in social participation
Uswatte; 2021 [21]—Birmingham	The outcome is the motor capacity	Built-in sensors and video cameras, participant opinion survey Participant opinion survey, motor activity log (MAL), The Wolf motor function test	Large improvements in everyday use of the more-affected arm	S730	The focus was on upper-extremity hemiparesis
Rozevink, 2021 [23]	Improvement of the upper limb motor ability quality of life, user satisfaction and motivation	Wolf Motor Function test (WMFT), arm function tests, the EuroQoL-5D-5L (EQ-5D), the intrinsic motiva- tion inventory (IMI), system usability scale (SUS) and Dutch–Quebec User	The WMFT, ARAT, and EQ-5D did not show significant differences 6 months after the training period when compared to directly after training. However, the FMA-UE results were significantly better at 6 months than at baseline	S730	Upper limb
Rozevink, 2021 [24]	Limb motor ability, quality of life	Wolf Motor Function Test (WMFT), action research arm test (ARAT), assessment upper extremity (FMA-UE), EuroQoL-5D (EQ-5D)	Progress in monitored game settings, user satisfaction and motivation	S730	Upper limb
Shih-Ching, 2021 [34]	Functional mobility, balance, and fall risk, the degree of perceived efficacy, classifying the strength in each of three lower extremity muscle actions (hip, gait)	Berg Balance Scale (BBS) scores, timed up and go (TUG) test, modified falls efficacy scale, Motricity Index, functional ambulation category	Improvement in balance	b730	Balance
Chingyi, 2021 [35]	Upper limb assessment, upper limb voluntary function, functional ability and motion speed of the upper limb, basic quality of participant’s ADLs, spasticity	The Fugel–Meyer assessment (FMA), action research arm test (ARAT), Wolf motor function test (WMFT), motor functional independence measure (FIM), modified Ashworth scale (MAS	Improvements in the entire upper limb	S730	Upper limb
Marin-Pard, 2021 [36]	EMG signal processing	Biofeedback, modular electromyography (EMG)	Development of a muscle-computer interface	S730	Upper limb function
Cramer; 2021 [21]—USA	Upper and lower lime function	Fugel–Meyer motor assessment	Assessments spanning numerous dimensions of stroke outcomes were successfully implemented	b730	Limb weakness
Kessler; 2021 [37]—Canada	Satisfaction of using telerehabilitation on the Client Satisfaction Scale (CSS)	Client Satisfaction Scale (CSS), Canadian Occupational Performance Measure (COPM)	High satisfaction and a strong therapeutic relationship	b730	Occupational performance coaching
Saywell, 2020 [25]	Physical function, hand grip strength and balance, self-efficacy, health outcomes	The physical subcomponent of the Stroke Impact Scale), A JAMAR hand-held dynamometer, the stroke self-efficacy questionnaire (SSEQ), overall stroke recovery rating of the SIS3.0	Rehabilitation augmented using readily accessible technology	b730	Physical function
Burgos; 2020 [38], Chile	Balance and functional independence user experience	BBS and Mini-BESTest (MBT), Barthel Index (BI), system usability scale (SUS)	Complementary low-cost telemedicine approach is feasible, and that it can significantly improve the balance of stroke patients	b730	Dosage and overall treatment
Ora; 2020 [22]—Norway	Feasibility and acceptability of speech and language therapy	Videoconference software called Cisco Jabber/Acano	Tolerable technical fault rates with high satisfaction among patients	b730	Post-stroke aphasia
Huzmeli; 2017 [12]—Turkey	Balance, Physical function, social role function, Emotional role function, mental health	The Berg Balance scale, short form-36 quality of life scale, The mini mental state	The balance levels significantly improved after the TR program, There was no difference in terms of quality of life and mental status before and after TR	b730	Post-stroke with hemiplegic
Ivanova; 2017 [39]—Germany	Motor relearning collection of instant feedback visualizations, incorporating telerehabilitation, arm motor gains, depression, pain, speed	Collection of instant feedback visualizations	Telehealth system for stroke rehabilitation using haptic therapeutic devices is currently being implemented into full functionality	b730	Stroke patients in recovering voluntary motor movement capability
Dodakian; 2017 [40]—USA	Incorporating telerehabilitation, arm motor gains, depression, pain, speed	Vital signs, magnetic resonance imaging, FM Scale, box and blocks (B&B), NIHSS, Barthel Index, geriatric depression scale (GDS) question form, mini-status exam (MMSE), optimization in primary and secondary control scale [20], Medical Outcomes Study Social Support Survey, Mental Adjustment to Stroke Scale (Fighting Spirit subscore), stroke-specific quality of life scale, modified functional reach forward displacement (cm), shoulder pain gait velocity stroke self-efficacy questionnaire	The results support the feasibility and utility of a home-based system to effectively deliver telerehabilitation	b730	Hemiparetic stroke
Özgün; 2017 [41]—Turkey	Cognitive levels, balance, quality of life	Mini Mental State Examination, Berg Balance Scale, short form- 36 (SF-36) quality of life scale	Improvement of using TR programs	b730	TR in patients with hemiplegia

## Data Availability

Not applicable.

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
