# Peer review of "Outcome Measures Utilized to Assess the Efficacy of Telerehabilitation for Post-Stroke Rehabilitation: A Scoping Review"

_brainsci, 2023, doi:10.3390/brainsci13121725_

Round 1

Reviewer 1 Report

Comments and Suggestions for Authors

The paper shows a number of important drawbacks as for the methodology and the references which I pointed out within the file I enclose. Good luck with the revision.

Comments on the Quality of English Language

none

Reviewer 2 Report

Comments and Suggestions for Authors

1. We recommend modifying the keywords to "post-stroke; dependence variable; telerehabilitation; rehabilitation assessment; telecare" to better align with the content of the paper.

2. We suggest reformatting the image layout of Figure 1 to ensure that the content  is presented completely and clearly to the readers.

3. Some sentences may lack clarity or contain ambiguity; we advise revising these sentences to ensure accurate communication of the intended meaning. For example, in the discussion section, the sentence "Importantly, the FMA has demonstrated outstanding reliability in both inter-rater and intra-rater assessments..." could be rephrased into several clearer sentences to elucidate the demonstrated reliability of FMA.

4. In the conclusion, you mentioned that future research should emphasize measuring the use and feasibility of these outcomes in the context of telerehabilitation (TR). We recommend further specifying the research directions in this regard. For instance, exploring comprehensive ways to assess intervention costs or investigating potential variations in TR acceptance among different demographic groups could be highlighted.

5. The paper mentions that future research should investigate the standards guiding clinicians and researchers in selecting outcomes. We suggest emphasizing the practical significance of this point in the conclusion. Discussing how these standards are crucial for improving the practical implementation and rehabilitation outcomes of telerehabilitation research can engage readers and stimulate further thought.

Comments on the Quality of English Language

Minor editing of English language required.

Reviewer 3 Report

Comments and Suggestions for Authors

The paper focuses on evaluating telerehabilitation (TR) in the context of post-stroke rehabilitation. The authors assess the literature related to TR for patients requiring post-stroke rehabilitation, surveying the outcome measures used in TR studies, and defining which parts of the International Organization of Functioning are measured in trials. 

1. The problem statement needs to be fleshed out even more. Why is this review timely and important?

2. The paper could benefit from recommending long-term follow-up studies to assess the sustained impact and effectiveness of TR interventions.

3. Exploring and including a wider range of technological modalities in TR could enhance the comprehensiveness of the research. This could include newer digital health technologies and emerging telecommunication methods.

4. Given the evolving nature of virtual reality (VR) in rehabilitation, a more detailed analysis of TR integrated with VR would be beneficial for understanding its specific impacts and effectiveness in post-stroke rehabilitation.

5. A section on general outlook and future directions on this topic should be written and is in fact the most important aspect. It needs to be emphasized even more.

Comments on the Quality of English Language

--

Round 2

Reviewer 1 Report

Comments and Suggestions for Authors

Dear Editor,

the paper improved after my comments. One more suggestion:

-provide the readers the search strings you adopted for all the databases reported (as supplementary material).

Best regards,

Comments on the Quality of English Language

Extensive editing of English language required

Reviewer 2 Report

Comments and Suggestions for Authors

The author responded well to my suggestion.

Author Response

thank you very much for your valuable comments.